# SMER28 Attenuates PI3K/mTOR Signaling by Direct Inhibition of PI3K p110 Delta

**DOI:** 10.3390/cells11101648

**Published:** 2022-05-16

**Authors:** Marco Kirchenwitz, Stephanie Stahnke, Silvia Prettin, Malgorzata Borowiak, Laura Menke, Christian Sieben, Carmen Birchmeier, Klemens Rottner, Theresia E. B. Stradal, Anika Steffen

**Affiliations:** 1Department of Cell Biology, Helmholtz Centre for Infection Research, 38124 Braunschweig, Germany; marco.kirchenwitz@helmholtz-hzi.de (M.K.); stephanie.stahnke@helmholtz-hzi.de (S.S.); silvia.prettin@helmholtz-hzi.de (S.P.); klemens.rottner@helmholtz-hzi.de (K.R.); 2Division of Molecular Cell Biology, Zoological Institute, Technische Universität Braunschweig, 38106 Braunschweig, Germany; 3Developmental Biology/Signal Transduction, Max Delbrueck Center for Molecular Medicine, 13125 Berlin, Germany; malgorzata.borowiak@amu.edu.pl (M.B.); cbirch@mdc-berlin.de (C.B.); 4Nanoscale Infection Biology Group, Helmholtz Centre for Infection Research, 38124 Braunschweig, Germany; laura.menke@helmholtz-hzi.de (L.M.); christian.sieben@helmholtz-hzi.de (C.S.)

**Keywords:** phosphatidylinositol 3-kinase (PI 3-kinase), mammalian target of rapamycin (mTOR), autophagy, receptor tyrosine kinase, small molecule, actin, hepatocyte growth factor/scatter factor (HGF/SF), platelet-derived growth factor-C (PDGF-C), cell proliferation, cancer biology

## Abstract

SMER28 (Small molecule enhancer of Rapamycin 28) is an autophagy-inducing compound functioning by a hitherto unknown mechanism. Here, we confirm its autophagy-inducing effect by assessing classical autophagy-related parameters. Interestingly, we also discovered several additional effects of SMER28, including growth retardation and reduced G1 to S phase progression. Most strikingly, SMER28 treatment led to a complete arrest of receptor tyrosine kinase signaling, and, consequently, growth factor-induced cell scattering and dorsal ruffle formation. This coincided with a dramatic reduction in phosphorylation patterns of PI3K downstream effectors. Consistently, SMER28 directly inhibited PI3Kδ and to a lesser extent p110γ. The biological relevance of our observations was underscored by SMER28 interfering with InlB-mediated host cell entry of *Listeria monocytogenes*, which requires signaling through the prominent receptor tyrosine kinase c-Met. This effect was signaling-specific, since entry of unrelated, gram-negative *Salmonella* Typhimurium was not inhibited. Lastly, in B cell lymphoma cells, which predominantly depend on tonic signaling through PI3Kδ, apoptosis upon SMER28 treatment is profound in comparison to non-hematopoietic cells. This indicates SMER28 as a possible drug candidate for the treatment of diseases that derive from aberrant PI3Kδ activity.

## 1. Introduction

Autophagy is an evolutionarily highly conserved cellular recycling process that is modulated by signals such as nutrient deprivation, metabolic stress, energy depletion and hypoxia. All these signals regulate the induction of autophagy, which also occurs on a basal level as a homeostatic process. Specific machinery controls this catabolic process in which double-membrane vesicles are formed to enclose various cytoplasmic constituents, such as protein aggregates, damaged organelles or intracellular pathogens. These double-membraned structures are termed autophagosomes and are finally delivered to and fuse with lysosomes to degrade and recycle the basic constituents [1,2,3]. A cellular kinase scaffold, termed the mTORC1 complex, is central to the regulation of autophagy and integrates signals on oxidative and energy stress as well as on the availability of, for example, growth factors or insulin to balance cellular growth. mTORC1 shares the protein subunits mTOR, Deptor and mLST8 with the mTORC2 complex, but is distinct from mTORC2 as it harbors the subunit RAPTOR. In addition, mTORC1 activity is selectively inhibited by rapamycin through the binding of a FKBP12-rapamycin complex. In contrast, mTORC2 retains its kinase activity upon acute rapamycin treatment, likely because the mTORC2-specific subunit RICTOR sterically prevents FKBP12-rapamycin binding [4]. A selective activator of mTORC1 activity is the small GTPase Rheb, which in turn is regulated by PI3K, AKT and the Rheb1-deactivating TSC complex. Thus, growth factor signaling through PI3K leads to active, GTP-bound Rheb1, thus, activating mTORC1 [1]. However, the enzymatic activity of PI3K is not only central to growth factor signaling but may also cause the malignant transformation of various cell types if the activity overshoots [5]. The etiology of hyperactivation can be due to copy number gain, increased expression or mutations leading to the pronounced or constitutive activation of PI3K isoforms, as well as the loss of antagonist function, such as of the lipid phosphatase PTEN [6]. Interestingly, a growing body of evidence shows that the PI3K/AKT pathway is interconnected with autophagy, suggesting that this might be a sensitive axis to interfere with cancer progression. In fact, a dual role of autophagy in the initiation and development of cancer currently constitutes an active area of research. On the one hand, autophagy is induced in different tumor types as a response mechanism to therapy. In line with this observation, the inhibition of autophagy was found to render cancer cells vulnerable for therapy [7]. On the other hand, the downregulation of autophagy by different means was shown to augment tumor development and, vice versa, the induction of autophagy conditions supports the defeat of tumors. More specifically, the deletion of several ATG (autophagy-related genes) and pro-autophagic kinases, such as AMBRA1 and Beclin, was observed to lead to the appearance of tumor lesions [8,9,10,11,12]. In addition, drugs known to target the PI3K/AKT/mTOR axis were reported to enhance radiation sensitivity in cancer cell lines and cancer models in pre-clinical studies [13,14,15]. Indeed, the screening of new target compounds and drug combinations already revealed anticancer effects by attenuating this PI3K/AKT/mTORC1 signaling axis [16,17,18].

SMER28 was identified in a screen for autophagy enhancing compounds acting independently of rapamycin [19]. While mTORC1 is clearly central to the regulation of autophagy, and rapamycin suppresses the kinase activity of mTORC1 through allosteric inhibition, the mode of action of SMER28 has remained elusive. Here, we reveal that SMER28 induces autophagy through a hitherto unknown mode of action. We show that SMER28 directly inhibits PI3K by binding to its catalytically active subunit p110. Moreover, we provide evidence that SMER28 causes growth retardation, accompanied by a partial arrest of the cell cycle in G1. SMER28 has little effect, however, on cell viability in osteosarcoma cells, as evidenced by cell cytotoxicity measurements. Interestingly, we find a higher susceptibility in B cell lymphoma, which show elevated apoptosis and necrosis rates. Attenuation of PI3K signaling by SMER28 is further confirmed by strongly reduced phosphorylation levels of Thr308-Akt and Ser473-Akt, two target sites of active PI3K. Consequently, SMER28 effectively blocked growth factor signaling downstream of receptor tyrosine kinases, as exemplified by the complete abolishment of cell scattering and dorsal ruffle formation elicited by HGF or PDGF. Together, here, we unveil the mechanism of autophagy induction by SMER28 through directly targeting the PI3K/AKT signaling axis, suggesting it as a promising lead structure for the development of anti-cancer drugs, most specifically B cell lymphomas due to its high affinity for the p110δ subunit.

## 2. Materials and Methods

### 2.1. Antibodies and Chemical Compounds

Antibodies, chemical compounds and other materials are detailed in the Appendix A.

### 2.2. Bacterial Strain and Cell Line Culture

*Salmonella enterica* serovar Typhimurium strain SL1344 was grown at 37 °C in Luria–Bertani (LB) medium under agitation. *Listeria monocytogenes* (EGD (wild-type)) and its isogenic internalin A and B-deleted strain (∆InlAB) were grown at 37 °C in brain heart infusion agar (BHI) medium under agitation.

U-2 OS human osteosarcoma cells, NIH/3T3 mouse embryonic fibroblast cells, c-Met control (WT/−) and KO (−/−) cells and Madin–Darby canine kidney (MDCK) cells were cultured in growth medium as follows: DMEM (4.5 g l^−1^ glucose) with 10% FCS, 2 mM L-glutamine, 1 mM sodium pyruvate and 1% non-essential amino acids. Murine B cell lymphoma WEHI-231 and A20 cells were maintained in RPMI 1640 with 10% FCS and 0.055 mM beta-mercaptoethanol. Cells were incubated at 37 °C in a humidified 7.5% CO_2_-atmosphere. C-Met control (genotype WT/−) and KO fibroblast cells were individually harvested from embryos of breeding with mice heterozygous for a c-Met null allele [20]. Upon genotyping, primary cultures with genotype WT/− as the control and −/− were immortalized by the retroviral transduction of SV40 LT antigen [21] using standard protocols.

### 2.3. Cell Lysate, Protein Measurements and Western Blotting

Protein extracts were prepared by harvesting cells that reached a confluency of approximately 70%. Cells were washed thrice with ice-cold PBS and lysed in ice-cold lysis-buffer (50 mM Tris, pH 7.5, 150 mM NaCl, 1 mM EDTA, 1% Triton-X 100) and SDS-Laemmli sample buffer as indicated in figure legends. The protein concentrations of cleared supernatants were determined using a BCA Protein Assay kit. Proteins were separated by SDS-PAGE and blotted according to standard procedures. Primary antibodies used in this study are listed in the key resource table. Signals were detected by using the Lumi-Light Western Blotting Substrate (Roche, Basel, Switzerland) with the ECL Chemocam IMAGER (Intas, Göttingen, Germany). Densitometric analyses of respective proteins were normalized to the levels of GAPDH, Tubulin or total protein, as indicated.

### 2.4. RT-qPCR

mRNA was extracted and stored at −80 °C prior to use. Real-time, quantitative PCR (RT-qPCR) was carried out with the SensiFAST SYBR No-ROX One-Step Kit in a LightCycler 96 (Roche, Basel, Switzerland) according to the manufacturer’s instructions. Primer details are listed in Appendix A. Reactions were incubated at 45 °C for 10 min and 95 °C for 2 min, followed by 45 cycles of 95 °C for 5 sec, 60 °C for 10 sec and 72 °C for 5 sec. The reference gene RPS9 was used for normalization. The 2^−∆∆CT^ method was used to calculate the relative changes in gene expression.

### 2.5. Cell Proliferation Assay

U-2 OS cells were seeded into a 24-well plate (3524, Corning, NY, USA) at a density of 20.000 cells/well and incubated for 16 h. Media were replaced with medium containing DMSO alone as the vehicle control or 50 µM or 200 µM SMER28 or 300 nM rapamycin. Cells were analyzed in an Incucyte S3 live-cell analysis system (Sartorius, Göttingen, Germany) with a 10× objective at 37 °C and 5% CO_2_. Four phase contrast images per well and four wells per treatment were acquired every hour for 47 h. The quantification of cell confluence was carried out with the Basic Analyzer of the Incucyte S3 and normalized to the average, respective initial confluence.

### 2.6. Annexin V Apoptosis and Necrosis Assay

U-2 OS cells were seeded at 3000 cells/well into 96-well plates (flat bottom) and treated as indicated for 48 h. After washing in annexin V-binding buffer (10 mM HEPES, 140 mM NaCl, 2.5 mM CaCl_2_, pH 7.4), cells were incubated for 15 min at RT in 50 µL annexin V-binding buffer containing propidium iodide (BioLegend, San Diego, CA, USA) and Alexa Fluor 488-conjugated annexin V (Thermo Fisher Scientific, Waltham, MA, USA). For the image-based analysis of U-2 OS cells, the adherent cell-by-cell module (IncuCyte S3) was used, allowing assessment of green and red fluorescence per each individual cell. For flow cytometric analysis of suspension B-cells, WEHI-231 and A20 were seeded at 100.000 cells/well into 96-well plates (round bottom) and treated as indicated for 24 h. Cells were washed once with Annexin V-binding buffer and incubated at RT in 50 µL Annexin V-binding buffer containing Alexa Fluor 488-conjugated Annexin V and Live/Dead Fixable Blue Dead Cell stain (Thermo Fisher Scientific, Waltham, MA, USA). Flow cytometric measurements were performed using the BD LSRII SORP system (BD Biosciences, Haryana, India) and data were analyzed with FlowJo software (BD Biosciences, Haryana, India).

### 2.7. Cell Viability Assay

U2-OS cells were seeded into white bottom 96 well plates at a density of 3000 cells/well, treated with DMSO alone as vehicle control or 50 µM or 200 µM SMER28 or 300 nM rapamycin, 150 nM epothilone B or 150 nM paclitaxel for 48 hours. Cell viability was assessed in a Cytation 5 plate reader (Agilent, Santa Clara, CA, USA) using CellTiter-Glo 2.0 Cell Viability assay (Promega, Fitchburg, WI, USA) according to manufacturer’s instructions. 

### 2.8. Cell Cycle Analysis

For cell cycle analysis, U-2 OS cells were treated for 24 h, as indicated. Cells were harvested by centrifugation and pellets washed twice with ice-cold PBS. Cells were fixed with 70% of ice-cold ethanol for 30 min followed by washing with PBS twice. 50 µg/mL RNase A and 50 µg/mL propidium iodide solution was added to the remaining cell suspensions. Subsequently, cell fluorescence was measured using the BD LSRII SORP system (BD Biosciences, Haryana, India). Fluorescence intensity analyses were performed with FlowJo software (BD Biosciences, Haryana, India).

### 2.9. Cell Scattering Assay

For scattering experiments, MDCK cells were seeded into a 24-well plate at a density of 3000 cells/well. Cells were stimulated after 32 h with 20 ng/mL HGF with or without 50 µM or 200 µM SMER28 as well as DMSO as the vehicle control. Cell scattering was imaged every 15 min for 12 h by phase contrast microscopy using a 10× objective at 37 °C in a humidified 7.5% CO_2_-atmosphere. For analysis of MDCK colony scattering after 12 h, scattered colonies were identified as cells that had lost contact to their neighboring cells.

### 2.10. Growth Factor Response Assay

For growth factor-induced membrane ruffling, 3 × 10^4^ NIH/3T3 cells were seeded onto fibronectin-coated glass coverslips in 24-well plates. Cells were washed with PBS three times and starved with DMEM containing respective inhibitors or vehicle control for 16 h. Then, cells were stimulated with 20 ng/mL HGF or 10 ng/mL PDGF in DMEM for 5 min as indicated. Coverslips were washed twice with cytoskeleton buffer (CB) (10 mM MES, 150 mM NaCl, 5 mM EGTA, 5 mM glucose, 5 mM MgCl_2_, pH 6.1) and fixed with pre-warmed 4% paraformaldehyde (PFA) in cytoskeleton buffer. Cells were permeabilized with 0.1% Triton X-100 in CB for 1 min and stained for the actin cytoskeleton with phalloidin Alexa Fluor 488 in CB for 1 h at room temperature. All coverslips were washed in CB and mounted with ProLong Diamond Antifade.

### 2.11. PI3 Kinase Inhibitor Assay

In vitro PI3 Kinase activity measurements were performed with the PI3K Kinase Activity/Inhibitor assay kit following the manufacturer’s instructions (Merck Millipore, Burlington, MA, USA). Briefly, kinase reactions were performed in provided glutathione-coated strips. Recombinant proteins of the isoforms of PI3K p110α, -β, -γ and -δ were each pre-incubated with the DMSO control, 50 µM and 200 µM SMER28 for 10 min. Kinase reaction buffer, PIP_2_ substrate and distilled H_2_O were added, followed by incubation at room temperature for 1 h. Biotinylated-PIP_3_/EDTA solution was added to all wells except the buffer control. Then, GRP1 solution was added and incubated for 1 h at room temperature. After washing four times with TBST, SA-HRP solution was added and incubated for one more hour. Next, wells were washed three times with TBST and twice with TBS before the addition of the substrate TMB solution. Reactions were incubated for 20 min in the dark and stopped by adding stop solution. Absorbance was read at 450 nm (Tecan Infinite 200pro; Tecan, Männedorf, Switzerland).

### 2.12. Gentamycin Protection Assay

For *Salmonella* Typhimurium and *Listeria monocytogenes* invasion assays, 5 × 10^4^ cells per well were seeded into 24-well plates 24 h before infection and incubated at 37 °C in a humidified, 7.5% CO_2_ atmosphere. Cells were pre-treated as indicated in figure legends. For *Salmonella* Typhimurium, LB broth was inoculated with an overnight culture (1:50) and grown at 37 °C under agitation up to an OD_600_ between 0.6 and 0.8. Bacteria were harvested by centrifugation at 3000× *g* for 3 min, and bacterial densities adjusted to a multiplicity of infection (MOI) of 100 in DMEM. Infections of respective cell lines were initiated after replacing media with pre-warmed DMEM, and the addition of *Salmonella* in DMEM immediately followed by centrifugation for 5 min at 935× *g*. After incubation for 30 min, 50 µg/mL gentamycin in DMEM was added for an additional 30 min.

For *Listeria* gentamycin protection assays, the overnight culture was directly harvested by centrifuging at 3000× *g* for 3 min, and bacterial densities adjusted to a MOI of 100 in DMEM. The infection of respective cell lines was performed essentially as described above, except that after incubation for 60 min, cells were washed once with PBS before the addition of 50 µg/mL gentamycin in DMEM for 2 h. Cells were then washed three times with PBS and lysed with 0.5% Triton X-100 in PBS for 5 min at room temperature. Lysates of all conditions were harvested on ice, and serial dilutions in PBS plated onto agar plates, followed by incubation at 37 °C overnight. Colonies were then counted with an automatic colony counter (Scan4000, Interscience; Saint Nom la Brétèche, France).

### 2.13. Planktonic Bacterial Growth Assay

Growth analysis of *L. monocytogenes* was performed with the Bioscreen C MBR (Turku, Finland). *L. monocytogenes* was grown at 37 °C in BHI broth overnight under agitation. OD_600_ was measured and set to 0.01 in BHI broth and fibroblast media supplemented with 50 µM SMER28, 200 µM SMER28 or 50 µg/mL gentamycin. OD_600_ was measured in quadruplicates every 15 min for 24 h during continuous shaking at 37 °C using EZExperiments.

### 2.14. Immunofluorescence

For immunofluorescence analysis, cells were seeded onto ethanol/acid-washed glass coverslips that were coated with 25 µg/mL fibronectin in PBS for 60 min at RT, essentially as previously described [22]. U-2 OS cells were cultured overnight before respective treatments. Cells were fixed with iced methanol (−20 °C) for 5 min followed by sequential rehydration in PBS. After PBS washing, cells were blocked with 5% horse serum in PBS containing 0.3% Triton X-100 for 30 min. Upon blocking, cells were incubated with primary antibodies diluted into 1% BSA in PBS for 1 h, washed with PBS and labeled with secondary antibodies for 40 min. After repetitive washing with PBS, cells were mounted with ProLong Diamond Antifade.

### 2.15. Image Acquisition

Fluorescence images shown in Figure 1A were acquired on a CSU-X1 (Yokogawa, Tokyo, Japan) spinning disk confocal microscope using a quad band filter connected to a Nikon Ti-Eclipse (Nikon, Tokyo, Japan) and a Modular Laser System 2.0 (Perkin Elmer, Waltham, MA, USA) equipped with 405, 488, 561 and 640 laser lines. Plan Apochromat immersion objectives (60× oil/1.4 NA and 100× oil/1.4 NA; Nikon) and an EMCCD C9100-02 camera (Hamamatsu, Hamamatsu City, Japan) were used. The system was controlled by the acquisition software Volocity 6.2.1 (Perkin Elmer, Waltham, MA, USA). Other spinning disk fluorescence images were acquired by a Nikon Ti2-Eclipse microscope with the spinning disk confocal module CSU-W1 (Yokogawa) using BP filters. Images were acquired with a Plan Fluor 40× oil/NA1.3 objective (Nikon), a Zyla 4.2 sCMOS camera (Andor, Belfast, UK) and 405/488/561/638 nm laser lines (Omicron, Rodgau-Dudenhofen, Germany) controlled by NIS-Elements software. For 3D structured illumination microscopy (SIM), cells were imaged with a Nikon Ti-Eclipse Nikon N-SIM E microscope and a CFI Apochromat TIRF 100× Oil/NA 1.49 objective (Nikon). Image acquisition was controlled by NIS-Elements software controlling an Orca flash 4.0 LT sCMOS camera (Hamamatsu, ), a Piezo z drive (Mad city labs, Madison, WI, USA), a LU-N3-SIM 488/561/640 laser unit (Nikon) and a motorized N-SIM quad band filter with a separate 525/50 emission filter using a 488 laser at 100% output power. Z-stack images were acquired with a step size of 200 nm. Reconstruction was carried out with the slice reconstruction tool (NIS-Elements, Nikon) using the reconstruction parameters IMC 2.11, HNS 0.37, OBS 0.07.

### 2.16. Data Processing and Statistical Analyses

Image analysis was carried out using ImageJ, Metamorph and NIS-Elements. Quantifications of LC3 and p62 structures were conducted by using the particle analyzer tool. Further data processing steps and statistical analyses were carried out in NIS-Elements, ImageJ, Inkscape, Excel 2010 and Graphpad Prism 9. For respective origin of resources, see Appendix A. Results from statistical tests, sample sizes and numbers of experiments are given in respective figure legends. Graphical illustrations were created with BioRender.com (accessed on 1 March 2022). Graphics were adapted from “PI3K/Akt, RAS/MAPK, JAK/STAT Signaling”, by BioRender.com (2022) and retrieved from https://app.biorender.com/biorender-templates (accessed on 1 March 2022).

## 3. Results

### 3.1. SMER28 Modestly Increases Autophagy

In order to document the effects of SMER28 on the regulation of autophagy marker proteins, we applied a series of analyses: We first assessed the appearance of autophagosomes in control and SMER28-treated U-2 OS cells by immunofluorescence with antibodies directed against LC3 and SQSTM1/p62 (Sequestosome-1; hereafter p62), two classical autophagosome markers [23]. Antibody staining revealed a higher number of LC3 and p62-positive autophagosomal puncta upon treatment with SMER28 (50 µM, Figure 1A). Quantification showed a comparable increase in numbers of LC3- and p62-positive puncta after SMER28 treatment (Figure 1B). Moreover, we measured the total areas of LC3 and p62 puncta per cell or per autophagosome (Figure 1C and Appendix A).

**Figure 1 cells-11-01648-f001:**
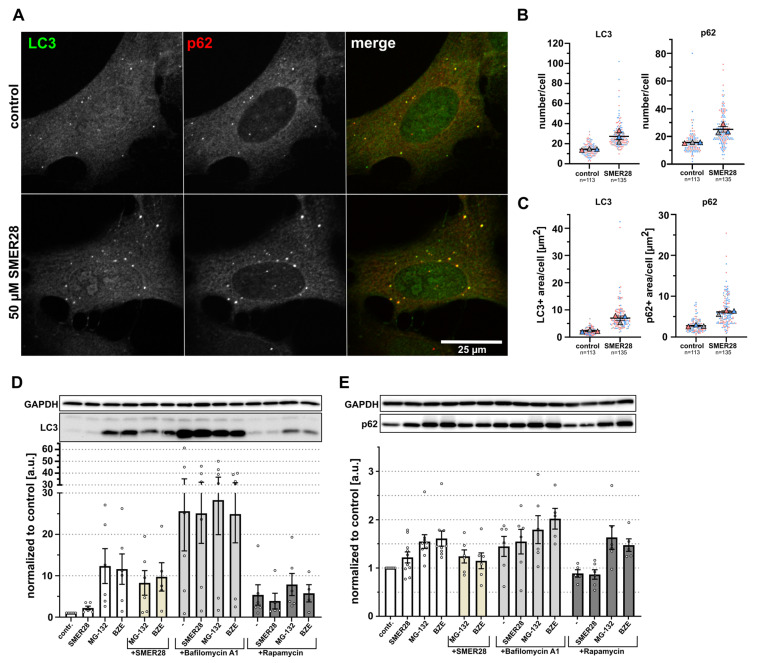
SMER28 stimulates autophagy in U-2 OS cells. (**A**) U-2 OS cells were left untreated (upper panel) or treated with 50 µM SMER28 for 16 h (lower panel). Cells were stained with p62/SQSTM1 and LC3 antibodies to assess markers for autophagosomes and visualized by confocal spinning disk microscopy. Merged images show LC3 in green and p62 in red. (**B**,**C**) Quantification of average numbers (**B**) and total areas (**C**) of LC3-positive and p62-positive puncta per cell. Data are shown as scatter plots with means ± s.e.m.; *n* = total number of cells analyzed; each biological replicate is colour-coded, large symbols depict means of respective experiments, according to “SuperPlots” [24]. (**D**,**E**) U-2 OS cells were treated with 50 µM SMER28, 1 µM MG-132, 15 nM bortezomib (BZE) and in the presence or absence of 250 nM bafilomycin A1 or 300 nM rapamycin for 16 h, as indicated and lysed in lysis-buffer. Protein levels of LC3 (**D**) and p62 (**E**) were assessed by WB; GAPDH was used as loading control. Graphs show quantifications of relative LC3-II and p62 levels after treatments as indicated. Data are means ± s.e.m; *n* ≥ 5.

We found that SMER28 treatment increased the areas of LC3 and p62 per cell approximately two- to threefold (Figure 1C), while the average sizes of autophagosomes in each case were only slightly augmented (Appendix A), indicating an increase in autophagosome numbers rather than an enlargement of these structures once formed. In addition, we found elevated levels of LC3-II, the lipidated form of LC3 [25], in SMER28-treated samples by Western blotting (Figure 1D), confirming earlier observations [19]. Surprisingly however, levels of p62 were also modestly increased after treatment with 50 µM SMER28 (Figure 1E). The induction of autophagy is usually connected to p62 degradation since it links polyubiquitinated proteins to the autophagosome and, hence, eventually to autophagic digestion. Thus, we investigated whether SMER28 might also have an inhibitory effect on the proteasomal pathway, which would explain the observed elevated p62 levels. To compare the effect of SMER28 treatment to well-established proteasomal inhibitors, we included bortezomib and MG-132 [26] in our analyses. While the treatment of cells with bortezomib and MG-132, respectively, led to an approximately 1.5-fold increase in p62 levels, SMER28 had a more modest effect (Figure 1E). Similarly, treatment with bortezomib or MG-132, respectively, increased LC3-II levels to a much higher extent than treatment with SMER28 (Figure 1D). The induction of autophagy by proteasomal inhibitors such as MG-132 has been described earlier [27,28]. Combining SMER28 with these proteasomal inhibitors resulted in lower LC3-II and p62 levels as compared to proteasome inhibition by MG-132 or bortezomib alone (Figure 1D,E). This suggests that SMER28 does not inhibit the proteasome, but instead may stimulate the turn-over of autophagy-related marker molecules. Interestingly, another, well-established autophagy-inducing drug, rapamycin [29], showed similar effects. Combined treatment of rapamycin together with MG-132 or bortezomib, respectively, reduced the average levels of LC3-II to extents similar to the combination of SMER28 with the above proteasome inhibitors (Figure 1D). To exclude that the observed increase in LC3-II and p62 upon SMER28 treatment was simply due to impaired lysosomal function, we compared the levels of LC3-II and p62 after SMER28 treatment in combination with bafilomycin A1, a specific v-ATPase inhibitor that inhibits the acidification of normally acidic organelles, such as lysosomes [30], inhibiting the degradation of the autophagosomal cargo. As expected, the endogenous LC3-II and p62 levels were substantially increased after co-treatment with SMER28 and bafilomycin A1 as compared to SMER28 treatment alone (Figure 1D,E), suggesting that SMER28 does not simply block lysosomal degradation. In an independent assay, we tested whether the application of SMER28 leads to an accumulation of polyubiquitinated proteins. While MG-132 and bortezomib clearly increased the levels of high molecular weight polyubiquitinated proteins, control and SMER28-treated cells showed indistinguishable levels of polyubiquitinated proteins (Appendix A).

### 3.2. SMER28 Treatment Leads to Growth Arrest

During our experiments, we noticed retarded cell growth in SMER28-treated conditions. To quantify this effect, we measured cell confluence of SMER28- and rapamycin-treated cells compared to controls over a time period of 47 h. We found that 50 µM SMER28 retarded cell growth to extents comparable to 300 nM rapamycin (Figure 2A,B). We also tested increasing concentrations of SMER28 (not shown) and found that 200 µM led to almost complete growth arrest after an initial lag phase of approximately 8 h, during which cell growth could still be observed (Figure 2A). In order to compare this cytostatic effect to well-established cytostatic drugs, we compared the effect of SMER28 and rapamycin to paclitaxel and epothilone B [31,32] on cell viability after 48 h. Low SMER28 concentration (50 µM) and 300 nM rapamycin had similarly low effects on cell viability, while a higher SMER28 concentration (200 µM) resulted in approximately 55% of viable cells, comparably in range to epothilone B and paclitaxel treatments (Figure 2B). To better distinguish between apoptosis and necrosis, we performed annexin V and PI (propidium iodide)-staining. Interestingly, more than 95% viable cells were observed in 50 µM SMER28 and 300 nM rapamycin treatment conditions, whereas 25% and 43% cell death was observed after 200 µM SMER28 and paclitaxel or epothilone B, respectively (Figure 2C). This effect was accompanied by an increased portion of apoptotic cells in case of SMER28, identified as annexin V-positive cells (Figure 2C). Altogether, the stronger growth inhibition upon increased SMER28 concentrations correlates, at least in part, with accelerated cell death.

These findings are comparable to the cytotoxic effects of paclitaxel and epothilone B, which are known to arrest the cell cycle at G2/M transition through the stabilization of microtubules [33]. Thus we wondered whether SMER28 could have similar effects and analyzed the cell cycle with different concentrations of SMER28- and epothilone B-treated cells by propidium iodide staining. Surprisingly, we found that the percentage of cells in G1 expanded with increasing SMER28 concentrations (Figure 3A,B), suggesting that SMER28 arrests cells, at least in part, at the restriction point. The cell cycle is tightly regulated by the activity of cyclin-dependent kinases (Cdks), which in turn are regulated by the abundance of their inhibitors and cyclins. Thus, we analyzed whether SMER28 affects the protein levels of aforementioned factors. CDK4/6 heterodimerize with D cyclins to form active kinase complexes allowing cell cycle progression from G1 to S, while the activity of these kinase complexes is inhibited by the association of p27/Kip and p21 Waf1/Cip1 with CDKs and cyclins through the formation of heterotrimeric complexes [34].

CDK4 and CDK6 levels were found to be largely unchanged in samples of control as well as SMER28-treated cells after 24 h (Figure 3C). Interestingly, however, both cyclins D1 and D3 showed decreased protein levels after 24 h of incubation with 50 µM and 200 µM SMER28 (Figure 3D), which is likely due to a decrease in mRNA levels (Appendix A). Furthermore, expression of the inhibitory subunit p27 was not affected by SMER28 treatment, while p21 levels decreased under SMER28 conditions (Figure 3E). This suggests that the activity of cyclins D1 and D3 is not sufficient for cell cycle progression from G1 to S, due to the diminished presence of these activating subunits cyclin D1 and D3.

### 3.3. SMER28 Interferes with Receptor Tyrosine Kinase Signaling

Next, we addressed which upstream regulatory pathway is targeted by SMER28 that may eventually lead to growth arrest. The signaling of growth factor receptor tyrosine kinases (RTKs) is well-established to be connected to proliferation as well as autophagy [35]. Thus, we tested whether SMER28 affected epithelial cell scattering induced by HGF (hepatocyte growth factor) by recording MDCK cells over a time period of 12 h of stimulation with 20 ng/mL HGF. DMSO control-treated MDCK cells scattered markedly from pre-formed colonies (Figure 4A, left panel, 4B; see also Supplementary Movie 1). In stark contrast, the application of 50 µM and 200 µM SMER28, respectively, completely abrogated the scattering of MDCK colonies (Figure 4A, middle and right panel, 4B; see also Supplementary Movie 1), indicating a strong defect in signal transduction downstream of the HGF/c-Met receptor. In a complementary approach, we recorded growth factor responsiveness by stimulating serum-starved NIH/3T3 cells with either HGF or PDGF (platelet-derived growth factor). F-actin staining revealed prominent responses in the control cells upon treatment with HGF or PDGF (Figure 4C, upper panel [36,37]).

Indeed, more than 50% of control cells responded with well-developed, dorsal ruffles upon treatment with either growth factor (Figure 4C,D). In contrast, less than 12% (50 µM) and 6% (200 µM) of SMER28-treated cells displayed dorsal ruffles upon HGF or PDGF stimulation, whereas the portion of cells with peripheral ruffles was nearly the same in all conditions (Figure 4C,D). Of note, most of the dorsal ruffles still formed in SMER28-treated cells were much smaller and/or not completely circularly closed. Collectively, these data reveal that SMER28 severely impairs the signaling downstream of RTKs, such as HGFR and PDFGR.

### 3.4. SMER28 Directly Targets PI3K Subunits Gamma and Delta

All findings described so far point towards a direct effect of SMER28 on the RTK/PI3K/mTOR signaling axis since we and others had observed an impact on autophagic flux (Figure 1) [19,38] and, in addition, on RTK signaling and growth retardation (see Figure 2, Figure 3 and Figure 4). The progression of cancer is frequently accompanied by aberrant kinase activities in this particular signaling pathway. To determine the molecular impact on this signaling cascade in more detail, we compared the effects of 50 µM and 200 µM SMER28 with rapamycin, a direct inhibitor of mTOR. Rapamycin affected phosphorylation of mTOR at Ser2448 as well as its downstream substrate p70S6K (p-Thr389) (Figure 5A,B). 50 µM SMER28 did not change phosphorylation levels of mTOR (p-Ser2448) after 4 h of treatment, confirming initial observations on rapamycin-independent effects of SMER28 [19]. Surprisingly, however, 200 µM SMER28 reduced the levels of phosphorylated mTOR to extents comparable to rapamycin (Figure 5A,B). This SMER28-dependent reduction in mTOR phosphorylation is also recapitulated by phosphorylation levels of p70S6K (p-Thr389) being reduced by approximately half (Figure 5A,B). While mTORC1 regulates various metabolic processes in cells, ULK1 is the first central downstream kinase specific for autophagy. One critical residue in ULK1 is Ser758 (Ser757 in mouse), which is phosphorylated by active mTORC1 to deactivate ULK1 [39]. After 4 h of treatment with 50 µM SMER28, we found a slight reduction of Ser758 phosphorylation of ULK1 (Figure 5A,B). Of note, 200 µM SMER28 treatment reduced the levels of p-Ser758 ULK1 by half, while rapamycin showed a less severe effect (28% reduction). This rather mild effect of rapamycin on the Ser758 phosphorylation site of ULK1 is known to be due to the low sensitivity of the mTORC1 phosphorylation site to rapamycin [40]. Together, the effects of SMER28 on these signaling kinases may account for the enhanced autophagy observed above (Figure 1) [19].

Since we had observed growth retardation through G1 cell cycle arrest and defective RTK signal transmission, we addressed whether SMER28 specifically affects kinase activities upstream of mTOR. Surprisingly, we found that SMER28 mildly reduced the levels of pTyr607 PI3K, whereas rapamycin did not change these levels, as expected (Figure 5A,B). RTKs also transmit signals through mitogen-activated kinases (MAPK) to regulate cell proliferation [41]. Interestingly, 200 µM SMER28 showed a noticeable decrease in the activation of MAPK as shown by reduced phosphorylation of pThr202/pTyr204 of MAPK (p44/42), while rapamycin did not affect this pattern (Figure 5A,B). The oncogene AKT is an indirect downstream effector of PI3K and phosphorylated by mTORC2 at Ser473 [42,43] and by PDK1 at Thr308 [44]. We, thus, quantitatively analyzed both phosphorylation sites of AKT and found, to our surprise, strongly reduced levels of Ser473-AKT phosphorylation by SMER28 (38% and 75% reduction with 50 µM and 200 µM SMER28, respectively) (Figure 5A,B). In contrast, rapamycin increased p-Ser473-AKT almost twofold (Figure 5A,B), which has been described to occur by mTORC1 inhibition [45,46]. These data, thus, corroborate the mTORC1-independent mode of action of SMER28 [19]. Interestingly, the levels of phosphorylated AKT at residue Thr308 were reduced by more than 50% by 200 µM SMER28, and to a much lesser extent with 50 µM SMER28 (Figure 5A,B), indicating that it acts on AKT phosphorylation indirectly and, thus, independent of its specific kinases mTORC2 and PDK1. Therefore, we tested whether SMER28 compromises PI3K function directly using an in vitro kinase assay. The active enzyme PI3K converts PIP2 to PIP3, which can be quantitatively assessed by adding biotinylated PIP3 (B-PIP3) as tracer. Class I PI3Ks comprise four different members that are differentiated by their catalytic subunits p110 alpha, beta, gamma and delta [5]. The ubiquitously expressed subunits alpha and beta were markedly inhibited with 200 µM SMER28, but only mildly affected with 50 µM of the compound. Notably however, PI3K activity exerted by the p110 delta subunit was eliminated with 200 µM SMER28 and almost entirely suppressed (by 87%) with 50 µM SMER28 (Figure 5C). The activity of the gamma subunit was inhibited more modestly, i.e., by 43% and 86% with 50 µM and 200 µM SMER28, respectively (Figure 5C). To confirm the mode of action of SMER28 in a cellular assay, and to assess the relevance of this drug in in vivo applications, we investigated the effect of SMER28 on *Listeria* invasion. *Listeria monocytogenes* is a gram-positive, bacterial pathogen that can cause abortions and illnesses, such as meningitis. *Listeria* invades host cells by binding of its surface and/or secreted factors, internalin A (InlA) and InlB to the host cell receptors cadherin-1/E-cadherin and c-Met/HGFR, respectively. As the InlA-E-cadherin interaction is restricted to the human system [47], internalin-specific entry into murine cells depends entirely on the InlB-Met pathway. Met/HGFR is widely expressed; therefore, signaling can be induced by the binding of its natural ligand HGF in various cell lines (see also Figure 4) as well as by the binding of bacterial InlB, known to involve PI3K activation [48,49,50,51]. We, therefore, addressed the invasion capability of *Listeria* in the presence and absence of SMER28 and compared this with the non-specific invasion assessed by *Listeria* lacking both InlA and –B (ΔInlA/B). Strikingly, 50 µM SMER28 reduced *Listeria* wild-type invasion by almost 90%, while 200 µM SMER28 completely abolished entry. Wortmannin and LY294002, two different pan-PI3K inhibitors, reduced *Listeria* invasion by only 39% and approximately 75%, respectively (Figure 6A,B). The differences of effects of both inhibitors could be due to the poor stability of wortmannin in solution [52]. Additional effects seen with these inhibitors on the invasion of Internalin A/B deletion mutants (*Listeria* ΔInlA/B) indicated a role for PI3K on non-specific, i.e., InlB-independent entry (Figure 6A,C). This conclusion was corroborated by comparing *Listeria* wild-type entry (with or without PI3K inhibition) into cells with or without genetic elimination of the HGF/InlB-receptor (c-Met). Results from c-Met-KO cells (Figure 6C) essentially phenocopied *Listeria* ΔInlA/B data in wild-type fibroblasts (Figure 6A,B). Moreover, wortmannin was again less effective in the control and c-Met KO cells than SMER28 at both concentrations employed (Figure 6C). These data demonstrated that SMER28 has robust effects on InlB-mediated *Listeria* invasion known to involve PI3K function in different cells lines. By contrast, gram-negative *Salmonella enterica* serovar Typhimurium, causing food-borne diarrhea, invade host cells largely independently of class I PI3 kinases [53,54]. We, thus, also explored the invasion capacity of *Salmonella* in the presence and absence of SMER28. Strikingly, SMER28 treatments did not significantly affect the invasion capacity of *Salmonella* into NIH/3T3 fibroblasts (Figure 6D), similar to what was found with LY294002 (Figure 6E).

To exclude that the compound has a harmful effect on *Listeria* itself, we performed growth curves of *Listeria* in bacterial growth and cell culture medium supplemented with 50 µM and 200 µM SMER28, respectively, or gentamycin as a classical antibiotic as the control, and compared these growth curves to the DMSO control. As shown in Figure 6F, none of the SMER28 conditions led to an adverse growth affect, while gentamycin effectively inhibited *Listeria* growth, with the exclusion that the reduced *Listeria* invasion rates observed above are due to the toxic effects of SMER28 on the bacteria.

### 3.5. SMER28 Significantly Affects Viability of B Cell Lymphoma Cells

Our data collected so far place the mechanism of action of SMER28 on PI3K activity, in particular, the p110 isoforms expressed in the hematopoietic system. Therefore, we sought to test the relevance of SMER28 for physiologically relevant B-cell malignancies, which are characterized by overactive PI3Kδ activity. Hence, we investigated the cytotoxicity of SMER28 on two different B cell lymphoma cell lines and compared these to cytostatic drugs affecting the microtubule cytoskeleton. Of note is WEHI-231, an established cell line for B cell lymphoma drug research, which was highly susceptible to 200 µM SMER28 (Figure 7A,B). Cell viability of WEHI-231 cells was affected by 50% after 24 h of treatment with 50 µM SMER28 (Figure 7A,B). Both epothilone B and paclitaxel also greatly reduced cell viability by approximately 75% and induced apoptosis (Figure 7A,B), indicating a high sensitivity of this cell line independently of the target. In contrast, the viability of A20 B cells remained largely unaffected by microtubule-stabilizing drugs after 24 h of treatment, whereas 200 µM SMER28 displayed effects similar to staurosporine, a non-specific protein kinase inhibitor. This also corresponded to more than 50% apoptotic or necrotic cells (Figure 7A,B).

Since we observed a greater impact on cell viability in B cell lymphoma as compared to osteosarcoma cells (Figure 2), we investigated signaling effects on the key components of PI3K signaling. In both B cell lymphoma cell lines, we found discernable effects on pSer473-AKT phosphorylation (Figure 7C,D). Consistent with the partially strong effect on cell viability and apoptosis (Figure 7A,B), we found a potent suppression of MAPK activation in WEHI-231 and A20 cells after treatment with 200 µM SMER28, as evidenced by the virtual absence of phosphorylated MAPK (Figure 7C,D). Moreover, 50 µM SMER28 decreased pThr202/Tyr204 levels of MAPK in WEHI-231 and A20 cells by more than 75%.

Hence, we propose to place the site of SMER28 action here in a model to the level of PI3K (Figure 8). Collectively, our data reveal a specific inhibition of class I PI3 kinases by SMER28 with differential activity towards respective p110 subunits, in particular, p110 delta.

## 4. Discussion

The initial reports describing an mTOR-independent, autophagy-inducing potential of SMER28 left an important question unanswered: what is the direct target of SMER28 mediating this activity [19]? While we were able to confirm a modest, autophagy-inducing effect of SMER28 here, we discovered, to our surprise, that SMER28 acts on the RTK signaling axis by directly inhibiting PI3K. More specifically, SMER28 exhibits its strongest inhibitory activity on functional class I PI3K complexes comprising the catalytic subunit p110 delta. It remains to be discovered whether SMER28 may have additional, biological targets. The enzymatic activity of PI3K is connected to essential cellular processes, such as cell growth, cell cycle regulation and cell migration [55]. In this study, we are able to connect several biological cell phenomena to SMER28-mediated in vitro inhibition of PI3K. We show that SMER28 has a cytostatic effect that is accompanied by low cytotoxicity in cells of non-hematopoietic origin, and cytotoxic effects on B cell lymphoma to some extent with concentrations investigated here, highlighting the role of PI3K as a growth factor/nutrient sensor [56]. We infer from these observations of different cell viabilities in divergent cell types—non-hematopoietic versus B cells—that the prime target of SMER28 is predominantly expressed in the more susceptible B cells. Moreover, we find that SMER28 has immediate effects on dynamic changes of the cytoskeleton, such as dorsal ruffle formation or long-lasting effects, such as cell scattering, both of which require cytoskeletal rearrangements [36]; such types of plasticity changes are key characteristics of cancer cells. Taken together, our findings disclose that SMER28 affects PI3K activity directly, explaining why the consequences of this treatment are more profound (e.g., see Figure 2 and Figure 7) than that of strict mTORC1 inhibition by Rapamycin [19], (see also Figure 8).

Aberrant PI3K signaling is commonly occurring in many types of cancers. Any type of misregulation of PI3K activity may lead to hyperactivation of the pathway, either by hyperactivating mutations of the enzyme itself, increased expression, increased PI3K copy numbers or loss of PI3K suppressors, such as its direct antagonist PTEN. This often contributes to the development of diseases, such as cancer, but also immunological or neurological disorders as well as diabetes [5,6,18,56]. Thus, the development of therapeutics for treatments based on PI3K has been driven forward with much effort. These activities resulted in the identification of a number of drugs that can be subdivided into dual PI3K/mTOR inhibitors, pan-PI3K inhibitors and isoform-specific inhibitors [16]. Class I PI3K comprises four different members that are differentiated by their catalytic subunit p110. PI3Kα and PI3Kβ are expressed ubiquitously, while PI3Kγ and PI3Kδ are enriched in the hematopoietic system [5]. B-cell malignancies, such as multiple myeloma (MM), chronic lymphocytic leukemia (CLL) and indolent Non-Hodgkin Lymphoma (iNHL), are strikingly dependent on PI3Kδ activity [56,57]. Hence, drugs specifically targeting this enzyme are in demand for the treatment of these lymphomas. The first PI3Kδ-specific inhibitor idelalisib (CAL-101) was approved by the FDA in 2014 [58,59,60]. Notably, idelalisib induces autophagy [57]. Unfortunately, idelalisib-treatment of relapsed CLL is accompanied by hepatotoxicity [61], which is hypothesized to be an off-target effect based on its structure. Subsequently developed, PI3Kδ-selective inhibitors, such as parsaclisib and umbralisib, were, thus, modified yet based on the chemical structure of idelalisib [62]. Although the chemical structure of SMER is known [19], the identification of the binding site of SMER28 to p110 delta and, ideally, the structural details of the binding will certainly help to understand the exact mechanism of inhibition. Moreover, it will be interesting to address whether the specific cytotoxicity of SMER28 towards selected tumor models as observed here could be translated into human application. Strikingly, the systemic treatment of mice with SMER28 [63] fitted the low cytotoxicity that had been reported earlier [64,65] and could be confirmed here, thus, constituting promising insight for future, organismic application. Specifically, our findings imply that SMER28 might be an applicable drug for treatment of B-cell non-Hodgkin-lymphoma. It is likely that SMER28 exerts multifaceted effects through its pivotal target PI3K, which is involved in manifold signaling pathways in addition to autophagy, such as microtubule stabilization [66,67]. Moreover, we cannot yet formally exclude kinases other than PI3K or other regulatory proteins as being affected by this compound directly. In addition to this, the PI3K signaling pathway emerges as the central switch for the effective infection of diverse pathogenic viruses, such as Ebola [68], West Nile virus [69] and Kaposi’s sarcoma herpesvirus [70]. Notably, APDS (activated PI3Kδ syndrome), an inherited immune disorder, is characterized by susceptibility to herpesviruses, such as Epstein–Barr Virus (EBV) [71]. Importantly, the inhibition of PI3Kδ is even discussed currently as a potential target to prevent SARS-CoV-2 infection through the suppression of the release of pro-inflammatory cytokines [72]. Further exploration of this hypothesis promises to provide important, potential knowledge for defeating infectious diseases in the future. Finally, our study also illustrates how different signaling pathways are intertwined with each other and sheds light on the connections between RTK signaling and autophagy.

## Figures and Tables

**Figure 2 cells-11-01648-f002:**
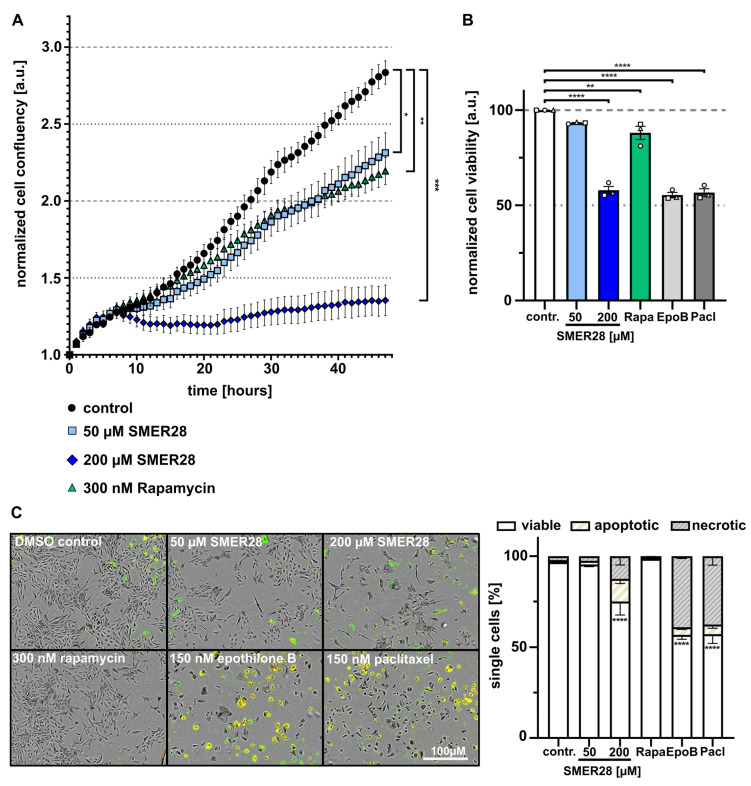
SMER28 treatment inhibits cell proliferation in a dose-dependent manner. (**A**) U-2 OS cells were treated as indicated and immediately recorded by phase contrast imaging for 47 h. The graph shows the mean confluence ± s.e.m. from at least four independent experiments with ≥4 replicates. * *p* < 0.05, ** *p* < 0.01, *** *p* < 0.001, two-way ANOVA. (**B**) Cell viability assay (CellTiter-Glo) of U-2 OS cells treated as indicated for 48 h. Data are means ± s.e.m; *n* = 3. ** *p* < 0.01, **** *p* < 0.0001, two-way ANOVA. (**C**) U-2 OS cells were treated for 48 h, as indicated, stained with propidium iodide (PI) and annexin V Alexa Fluor 488 and imaged by phase contrast and fluorescence microscopy. Images show overlays after 48 h, the graph shows quantifications. Cells were categorized as viable versus apoptotic (green fluorescence-positive) or necrotic (green and red fluorescence-positive). Data were collected from three independent experiments and represent means ± s.e.m. **** *p* < 0.0001, two-way ANOVA.

**Figure 3 cells-11-01648-f003:**
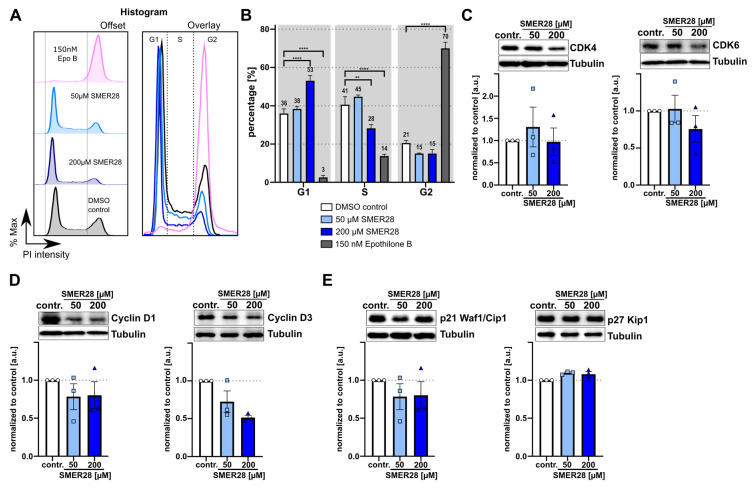
SMER28 treatment induces G1 cell cycle arrest. (**A**,**B**) Flow cytometry analysis of cell cycle distributions using PI (propidium iodide) staining after 24 h incubation of U-2 OS cells with different concentrations of SMER28 or epothilone B, as indicated. (**A**) Representative histograms depicting the distributions of cell cycle phases for indicated treatments. Histograms are shown as offset (left) and overlay (right). (**B**) Bar plots showing respective percentages of cell cycle phases (G1, S and G2) according to PI intensities. Percentage of each cell cycle phase was calculated by Watson Pragmatic model using FlowJo software. Data are means from at least three independent experiments ± s.e.m. ** *p* < 0.01, **** *p* < 0.0001, two-way ANOVA. (**C**–**E**) U-2 OS cells treated with 50 or 200 µM of SMER28 were lysed in Laemmli buffer after 24 h. Protein expression levels of CDK4 and CDK6 (**C**), Cyclin D1 and Cyclin D3 (**D**) as well as p21/Cip1 and p27/Kip1 (**E**) were assessed by WB. Bar graphs show protein levels calculated relative to tubulin. Data are means ± s.e.m; *n* = 3.

**Figure 4 cells-11-01648-f004:**
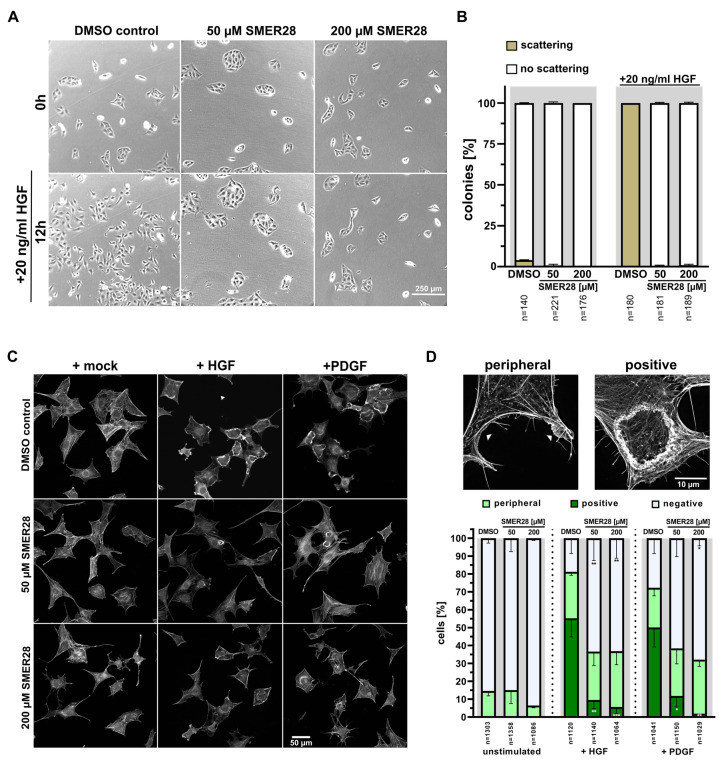
SMER28 inhibits growth factor-mediated cell scattering and membrane ruffling. (**A**) Colony-forming MDCK cells were stimulated with 20 ng/mL HGF in the presence or absence of 50 or 200 µM SMER28, as indicated, and imaged every 15 min by phase contrast microscopy for 12 h. (**B**) Quantification of MDCK scatter assay. Colonies were categorized as “no scattering” versus “scattering” as described in Materials and Methods. Data were collected from three independent experiments and represent means ± s.e.m; *n* = total numbers of cells analyzed. (**C**) NIH/3T3 cells were serum-starved for 16 h in DMEM with treatments as indicated on the left. Cells were then treated with DMEM with or without HGF or PDGF for 5 min, as indicated, stained with phalloidin to visualize the F-actin cytoskeleton and imaged by spinning disk microscopy. (**D**) Quantification of growth factor responses according to examples shown above the graph. Images show maximum intensity projections of super-resolution SIM images. Cells were categorized as being non-responsive (negative) versus harboring peripheral ruffles (peripheral) or circular dorsal ruffles (positive). Data were collected from three independent experiments and represent means ± s.e.m; n=total number of cells analyzed. * *p <* 0.05, ** *p <* 0.01, *** *p* < 0.001, one way ANOVA.

**Figure 5 cells-11-01648-f005:**
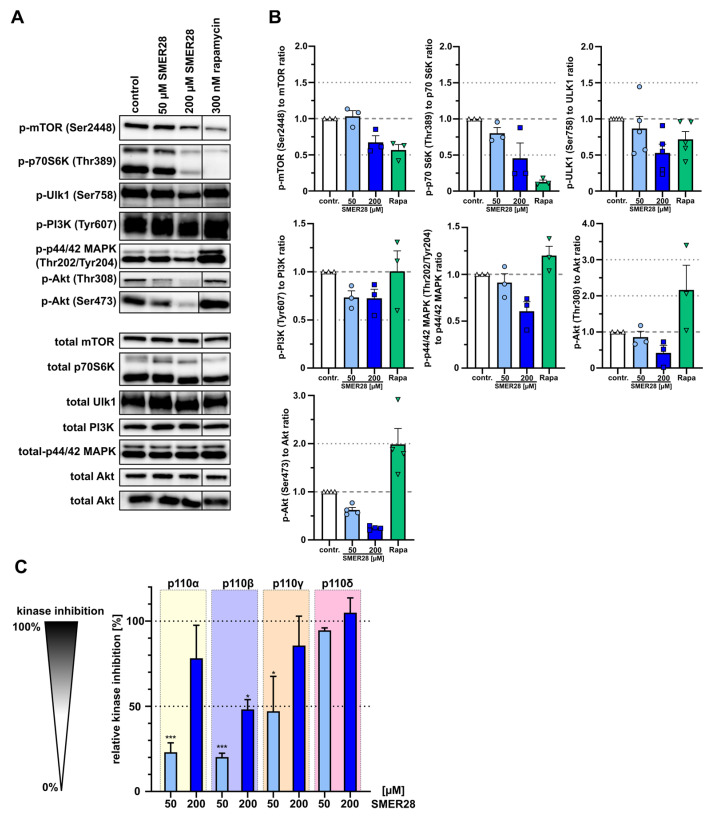
SMER28 inhibits PI3K signaling. (**A**) Western Blot analysis of U-2 OS cells. Cells were incubated in the presence or absence of 50 or 200 µM of SMER28 or 300 nM rapamycin (Rapa), as indicated, for 4 h, lysed in Laemmli buffer and analyzed for levels of phospho-proteins and their respective total proteins. (**B**) Quantifications of relative ratios of phosphorylated proteins to total proteins as indicated. Data are means ± s.e.m. of protein levels derived from at least three independent experiments. (**C**) Inhibition of activities of class I PI3 kinases containing the regulatory subunits p110α, -β, -γ and -δ by 50 or 200 µM SMER28, as measured using an in vitro PI3 kinase activity/inhibitor ELISA. The respective relative kinase inhibition was normalized to the data of the reaction without kinase, equaling 100% inhibition per definition. Data represent mean values ± s.e.m; *n* = 4. * *p* < 0.05, *** *p* < 0.001, compared to the reaction without kinase (not shown, 100%); one way ANOVA.

**Figure 6 cells-11-01648-f006:**
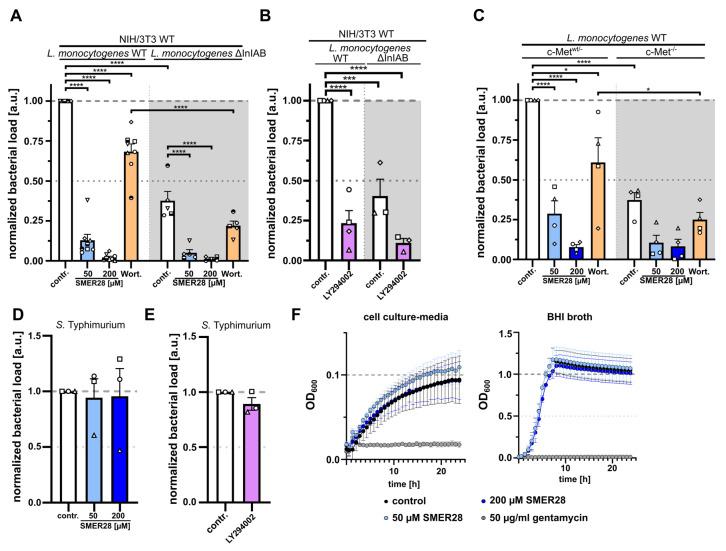
SMER28 efficiently inhibits *Listeria* uptake. (**A**) NIH/3T3 cells pre-treated with or without 50 or 200 µM SMER28 or 50 nM wortmannin for 16 h were explored by gentamycin protection assay for invasion of *L. monocytogenes* wild-type and ∆InIA/B mutant. (**B**) NIH/3T3 cells were pre-treated with or without 50 µM LY294002 for 30 min and subjected to gentamycin protection assay using *L. monocytogenes* wild-type and ∆InIA/B. (**C**) c-Met^WT/−^ and c-Met^−/−^ cells were pre-treated as described in A and examined by gentamycin protection assay with *L. monocytogenes* wild-type as above. (**D**) NIH/3T3 cells were pre-treated with or without 50 or 200 µM SMER28 for 16 h. Cells were then subjected to gentamycin protection assay with *S.* Typhimurium as invasive pathogen. (**E**) NIH/3T3 cells were pre-treated with or without 50 µM LY294002 for 30 min and examined by gentamycin protection assay for invasion of *S.* Typhimurium. In (**A**–**E**), all bars display means from at least three independent experiments ± s.e.m. (**F**) Planctonic growth assay of *L. monocytogenes* wild-type bacteria in cell culture media (left graph) and in BHI broth (right graph). Media were supplemented with or without 50 µM SMER28, 200 µM SMER28 or 50 µg/mL gentamycin, as indicated. OD600 was measured every 15 min for 24 h. Graphs display means from three independent experiments ± s.e.m. * *p* < 0.05, *** *p* < 0.001, **** *p* < 0.0001; one way ANOVA.

**Figure 7 cells-11-01648-f007:**
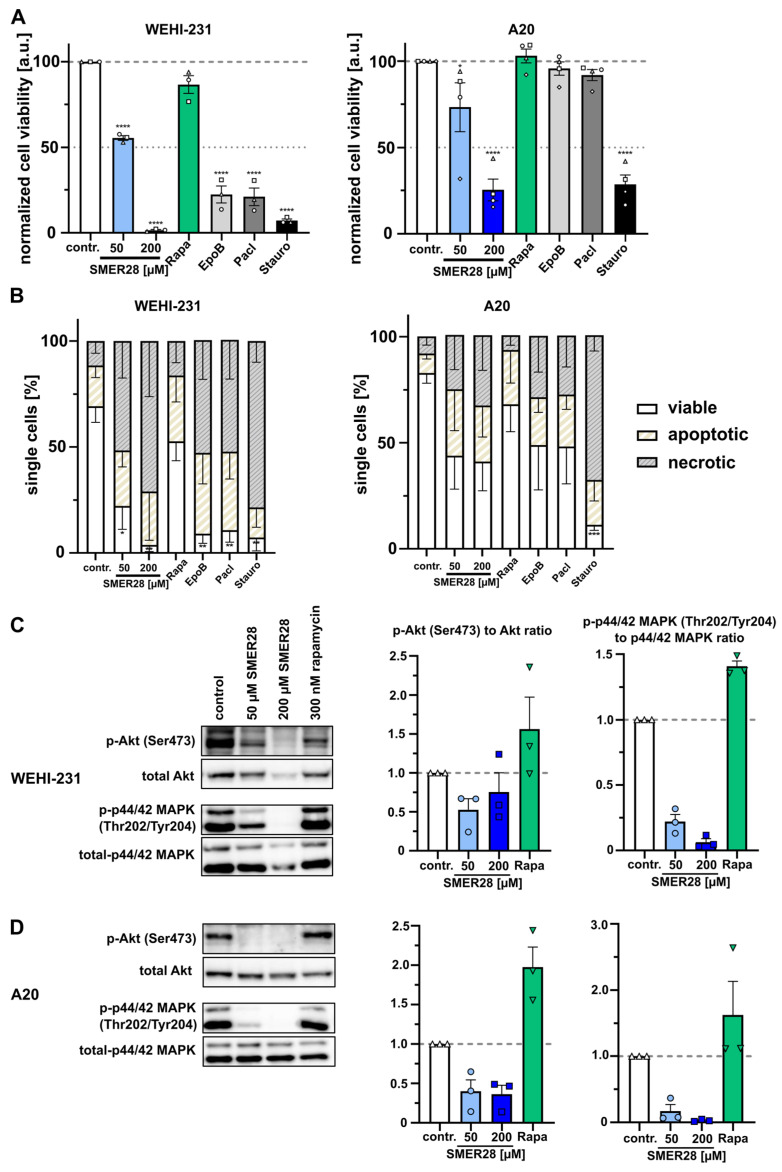
SMER28 reduces B cell viability to different extents. (**A**) Cell viability assay (CellTiter-Glo) of WEHI-231 and A20 cells treated as indicated for 24 h. Graphs show mean cell viability. Data are means ± s.e.m; *n* = 3. * *p* < 0.1, **** *p* < 0.0001, two-way ANOVA. (**B**) Cells were treated as indicated for 24 h, stained with Live/Dead stain and annexin V Alexa Fluor 488 and analyzed by flow cytometry. Cells were categorized as viable versus apoptotic or necrotic. Data were collected from three independent experiments and represent means ± s.e.m. * *p* < 0.1, ** *p* < 0.01, *** *p* < 0.001, **** *p* < 0.0001, two-way ANOVA. Western blot analysis of WEHI-231 (**C**) and A20 (**D**) cells. Cells were incubated in the presence or absence of 50 or 200 µM of SMER28 or 300 nM rapamycin, as indicated, for 4 h, lysed in Laemmli buffer and analyzed for levels of phospho-proteins and their respective total proteins. Data are means ± s.e.m. of protein levels derived from three independent experiments.

**Figure 8 cells-11-01648-f008:**
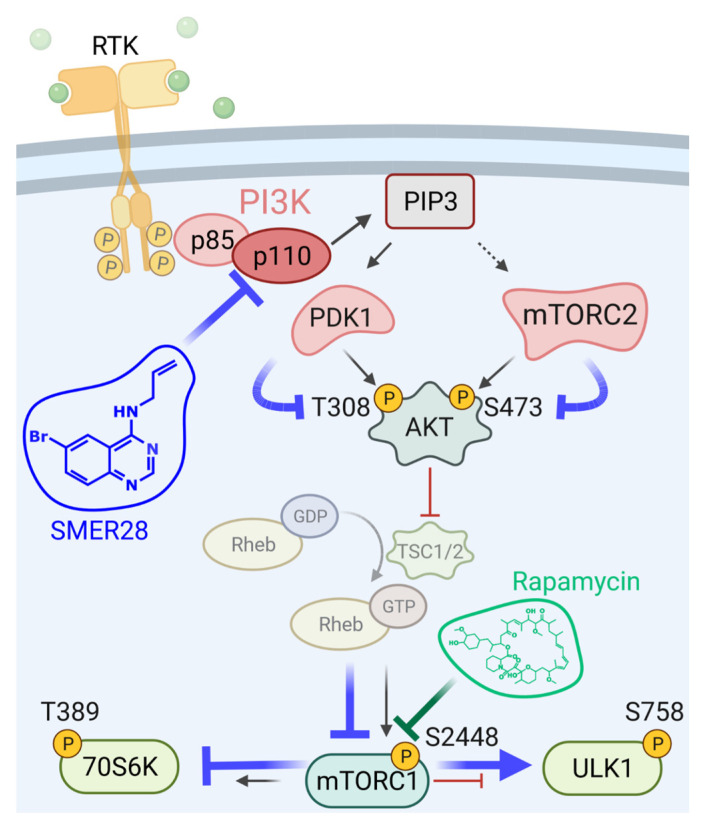
Scheme proposing the place of action of SMER28. Blue lines indicate the inhibitory (T) or stimulatory (↑) activity of SMER28.

## Data Availability

Not applicable.

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
