# Peer review of "SMER28 Attenuates PI3K/mTOR Signaling by Direct Inhibition of PI3K p110 Delta"

_cells, 2022, doi:10.3390/cells11101648_

Round 1

Reviewer 1 Report

The Authors’ aim is to explore the mechanisms of action of SMER28 (Small molecule enhancer of Rapamycin 28), an autophagy-inducing compound that acts via mTOR-independent mechanism(s). They therefore assess the effect of the compound on the cell cycle/cell proliferation and on the signaling pathways activated downstream of the RTKs. The initial idea is good but sometimes the experiments are performed in different models thus impeding to understand whether the MOA might be general or just related to a certain stimulus in a certain model. Given that in more than one point in the manuscript the Authors suggest that SMER28 might be useful in B cell lymphoma they should investigate its effect in a relevant model of B cell neoplasia. However, given the interesting effects on MET-driven signaling they should also explore the effects of the compound in a relative MET-dependent tumoral model. Finally, the manuscript suffer of a lack of biochemical assays which better define – compared to the microscopy-based ones -  the quantitative aspect when studying the effect of the compound on different cell-related processes. I therefore strongly encourage the Authors to add results obtained with more standard biochemical assays.

In conclusion, a number of experiments should be added to corroborate the present data and make the manuscript acceptable for the publication

Major points:

  • 2 SMER28 treatment inhibits cell proliferation in a dose-dependent manner.

Rather than using an imaging system a more accurate estimate of cell death and inhibition of cell growth is given by biochemical assay such as the ATP assay (Cell-Titer-Glo, Promega) and the BrdU incorporation assay (Cell Proliferation ELISA, BrdU, colorimetric frm Sigma-Aldrich), respectively.

  • Line 336: This suggests that the overabundance of p27 during the concomitant lack of the activating subunits cyclin D1 and D3 may lead to the observed partial arrest in G1 to S phase.

Since p27 levels are unchanged there is no overabundance. It’s the decrease in CyD1 and D3 levels that likely impedes the progression beyond the G1, not the presence of too much inhibitor but the lack of the activity needed for the progression. Please rephrase the sentence. In addition it would be interesting to assess whether the decrease in cyc D1 and D3 is due to a lack of transcription/translation or an increase in the degradation. That would help in the understanding of SMER28 MOA.

Line 394 Collectively, these data reveal that SMER28 severely impairs signaling downstream of RTKs such as HGF and PDFG.

Given this conclusion it would be interesting to see in which types of epithelial tumors the compound is effective. In addition, since the Authors eventually suggest that this compound might be useful in the treatment of  B cell lymphomas why did they investigate only the effect on growth factors that usually regulate the physiology of epithelial cells? Experiments performed in B cell neoplasia would be more informative, given that they respond to different stimuli and growth factors and depend on the activation of different receptors for growth and survival, such as the BCR. The Authors should repeat the experiments addressed to understand to MOA of SMER28 in more appropriate models of B cell malignancies. In addition, to fully understand which downstream pathways are affected by SMER28 they should check the RAS/MAPKs not only the PI3K/AKT since it is known that PI3K can be activated directly downstream of RAS (Downward, J. Nat Med 14, 1315–1316 , 2008)

Reviewer 2 Report

The paper by Kirchenwitz et al entitled "SMER28 attenuates PI3K/mTOR signaling by direct inhibition of PI3K p110 delta" is an interesting study in which the authors demonstrate that SMER28  affects PI3K activity directly, other than to induce several biological effects .

The authors produces a well organisezed experimental design, the results are  claerly presented and in my opinion the manuscript is suitable for the publication.

Reviewer 3 Report

Even though SMER28 was identified in a screen of autophagy enhancing compounds acting independently of rapamycin (Ref. 19). In this study, the SMER28 is an autophagy-inducing compound functioning by assessing classical autophagy-related parameters. They discovered several additional effects of SMER28, including growth retardation and reduced G1 to S phase progression. Most strikingly, SMER28 treatment led to a complete arrest of receptor tyrosine kinase signaling, and consequently growth factor-induced cell scattering and dorsal ruffle formation. The data also reveal a specific inhibition of class I PI3 kinases by SMER28 with differential activity towards respective p110 subunits, in particular p110 delta.

Overall, these findings are significant and sufficient to be deserved for publication.

However, one concern is that the concentration of SMER28 (50-200uM) used in Figures 3 is still higher than Baf (250 nM=0,25uM), Rapmycin (300nM=0.3uM), and two Proteosome inhibitor: 1 µM MG-132, and 0.015 uM bortezomib (BZE). Another concern is about final cells fate, I would like to know Annexin V staining profile to show cell fate (such as apoptosis) in Figure 3 when combinations SMER28 (50 or 200uM) with Baf, Rapmycin, MG-132 and bortezomib (BZE)

Round 2

Reviewer 1 Report

The Authors satisfied the issues I raised, therefore i recommend the paper for publication